# Extraperitoneal Laparoscopic Approach in Inguinal Hernia—The Ideal Solution?

**DOI:** 10.3390/jcm11195652

**Published:** 2022-09-25

**Authors:** Bogdan Barta, Marina Dumitraș, Ștefana Bucur, Camelia Giuroiu, Raluca Zlotea, Maria-Magdalena Constantin, Victor Mădan, Traian Constantin, Cristina Raluca Iorga

**Affiliations:** 1General Surgery Clinic, Euroclinic Regina Maria Hospital, 070000 Bucharest, Romania; 2Faculty of Medicine, “Carol Davila” University of Medicine and Pharmacy, 050474 Bucharest, Romania; 32nd Department of Dermatology, Colentina Clinical Hospital, 020125 Bucharest, Romania; 4Department of Urology, Emergency University Central Military Hospital, 010825 Bucharest, Romania; 5Department of Urology, “Prof. Dr. Th. Burghele” Hospital, 050652 Bucharest, Romania; 6Surgery Clinic, “Dr. Carol Davila” Clinical Nephrology Hospital, 010731 Bucharest, Romania

**Keywords:** TEP, TAPP, inguinal hernia, Lichtenstein, laparoscopy

## Abstract

Background: After more than 20 years since laparoscopy was proposed as a solution for one of the most common surgical pathologies, inguinal hernia, the choice of an intra- or extraperitoneal approach has remained a highly debated topic. Purpose and objectives: This study aimed at analyzing the feasibility of the extraperitoneal approach, by routine for this team/ and answering the question of whether this type of approach can be considered a safe one. Although indications for an intra- or extraperitoneal approach largely overlap, it may also be a matter of surgeon preference in choosing one technique. Methods: The study was retrospective, conducted on a group of 493 patients operated on for inguinal hernia in the clinic, by a single operating team, between January 2012 and March 2022. Results: It was proven that out of the 493 surgeries for inguinal hernia, 95.1% (*n* = 469) were operated upon by laparoscopic TEP (total extra peritoneal patch plasty approach); 1.62% (*n* = 8) by laparoscopic TAPP (transabdominal intraperitoneal); and 3.24% (*n* = 16) by the open, anterior approach (Lichtenstein). There were no intraoperative complications recorded in any of the procedures, while postoperative complications were found in 10.23% of cases (*n* = 48) in the extraperitoneal approach, and recurrences after the TEP approach were recorded in 0.40% of cases (*n* = 2). Conclusions: For correctly selected cases, TEP hernia surgery can be considered a safe and reliable approach.

## 1. Introduction

Surgical repair of inguinal hernia is the second most common intervention in general surgery. In the United States, there are 800,000 inguinal hernias operated on annually, and 20 million worldwide [1,2,3].

After more than 20 years since laparoscopy was proposed as a solution for one of the most common surgical pathologies, inguinal hernia, the choice of a transabdominal intra- peritoneal (TAPP) or total extraperitoneal (TEP) approach has remained a highly debated topic [4,5].

Most studies have shown the effectiveness and safety of the two types of approach without being able to identify the superiority of any of them. Data from the literature are unanimous regarding the benefits of the laparoscopic approach in the treatment of hernias, such as rapid recovery, reduced postoperative pain, low incidence of urinary retention and low risk of suppurative complications, short hospitalization (one day), and rapid socio-professional reintegration (one week) [2,6,7]. In the TEP technique, the additional benefits are represented by the avoidance of injury to the intra-abdominal organs and the formation of adhesions. The shortcomings of the method are related to a higher risk of damage to the vessels: the epigastric pedicle, the funicular elements, and a longer learning curve [1,8,9]. The intraperitoneal onlay mesh repair (IPOM) procedure has been abandoned [9,10].

On the basis of the surgical experience of the clinic in hernia pathology, this study aimed to analyze the feasibility of the extraperitoneal approach, specifying the indications and limitations of the method [3,11].

## 2. Materials and Methods

The retrospective study was conducted on patients operated for inguinal hernia between January 2012 and March 2022. It included 493 consecutive patients operated on in our clinic by a single operating team, with experience in laparoscopic hernia repair (10–15 years of experience in this field). The study included patients operated upon laparoscopically, by first intention, according to the inclusion criteria, on the basis of parameters such as age, gender, type of hernia—unilateral/bilateral, length of hospitalization, indications or contraindications for a certain method, intra- or postoperative complications, features related to surgical procedure (duration of the operation), drainage, conversion (laparoscopy—open) or relapse/recurrence rates, postoperative control (routine) at 1.5 weeks (stich removal, clinical aspect of the wound), and correlations between these parameters [1,12,13] (Table 1).

For statistical data analysis, we used IBM SPSS statistics 21.

The inclusion criteria for the TEP group were patients with reducible hernia, incarcerated hernia, recurrent hernia after classic procedure, ASA score 1–3, history of intra- abdominal infections, and no previous surgical interventions in the lower abdomen.

TEP required the use of three trocars placed as standard, on the basis of the principle of triangulation: 5 mm trocar on the midline, halfway between the umbilical–pubic distance, under optical control (10 mm trocar inserted in the sub-umbilical region), and another 5 mm trocar inserted at 3–4 cm superior and one at 1–2 cm medial of the antero-superior spine. The mesh used consisted of a polypropylene/Bard 3D Max Light Mesh—SorbaFix, Bard Absorbable Fixation System. Redon drains were in particular required in bilateral interventions with extensive dissection and with a higher hemorrhagic risk (patients on antiplatelet therapy, intraoperative bleeding) [1,3].

The inclusion criteria for the TAPP group were reducible hernia, recurrent hernia after classic procedure, patients with ASA score 1–3, irreducible or incarcerated hernia with intestinal loop in the hernia sac, and patients with previous subumbilical surgical interventions.

As seen above, the most important criteria for differentiating the two groups were the history of previous interventions with subumbilical scar and the presence of an intestinal loop blocked in the hernia sac (in these cases, we preferred TAPP).

TAPP technique requires three trocars, a 10 mm optic trocar placed 0.5–1 cm below the umbilicus and two 5 mm trocars placed at the level of the umbilicus at the outer edge of the rectus abdominis on the hernia side and below the level of the umbilicus at the outer edge of the rectus abdominis on the healthy side. After inspection of the abdominal cavity (type and classification of the hernia, detection of occult hernia on the other side), the peritoneum was cut in an arc at about 2 cm above the hernia ring from the medial umbilical ligament to the anterior superior iliac spine. The medial pubic bladder space and the lateral iliac fossa space were dissected, and the hernia sac was exposed and separated from the spermatic cord elements. A synthetic mesh was inserted to completely cover the whole myopectineal orifice. The peritoneum was closed using continuous suture or AbsorbaTack.

The classic Lichtenstein procedure has been indicated only in cases of large, complicated hernias, in patients with coagulation disorders with severe cardiac or respiratory impairment when the pneumoperitoneum is contraindicated. In these cases, we preferred to insert a drain tube (almost a routine) as there were patients with high hemorrhagic risk with extensive dissection and risk of suppuration.

The open approach was represented by the Lichtenstein “tension-free” procedure.

The Lichtenstein technique consists of the incision of the skin in the inguinal area, the dissection of the subcutaneous tissue, the incision of the external oblique aponeurosis, and the isolation of the spermatic cord. In the case of indirect hernias, the hernia sac is identified, dissected to the internal ring, and opened in order to examine its content. Usually, the sac is ligated, and the distal part is excised. In the case of direct hernia, we prefer to imbricate its content using non-absorbable sutures. Then, a polypropylene mesh is used to strengthen the posterior wall of the inguinal canal and it is sutured to the fibromuscular structures in a tension-free manner. After meticulous hemostasis, a suction drain is placed in particular in large inguinal hernias with extensive dissection. The oblique aponeurosis and the surgical incision are sutured.

The laparoscopic approach consists of TAPP or TEP techniques.

The diagnostic protocol consists of (1) clinical examination: diagnosis of uni- or bilateral hernia made during a clinical examination of hernia points; (2) soft tissue ultra- sound: used and recommended when the diagnosis is not readily apparent, when we detect a small hernia point, or when we unilaterally highlight a deeply enlarged inguinal ring to confirm/deny the existence of a contralateral inguinal hernia. CT or MRI scans are rarely required. We inserted a drain tube after interventions with extensive dissection (bilateral hernias), or in the case of patients with increased hemorrhagic risk (e.g., on antiplatelet therapy).

## 3. Results

The age of the patients included in the study was between 25 and 86 years (Table 2); the average age of TEP patients was 48 years compared to 75 years, the average age of those with open surgery.

In the case of patients in the TEP and TAPP group, the mean and median had almost similar values, but in the case of the open surgery group, we had different values: the mean of 68 and the median of 83. This proves that the Lichtenstein procedure was indicated in the cases of elderly patients, who also presented multiple comorbidities.

The two groups of patients operated by the TAPP and classic procedures, not being statistically significant, cannot be compared with the group of patients operated with TEP. These procedures were performed when the limits of the TEP were exceeded.

Furthermore, 31% (*n* = 152) of the total of 493 operated patients were elderly patients (over 60 years old), and 84.2% of patients over 60 years old (*n* = 128) were operated with TEP; therefore, we can conclude that the TEP procedure is also feasible for well-selected elderly patients (Table 3).

Regarding the distribution by gender, the majority of operated patients, 83.97%, were men (*n* = 414 patients) who underwent TEP surgery (Table 4).

As for the type of hernia, TEP was performed in most cases for reducible, unilateral, and bilateral hernias in 53% of cases (*n* = 249); after relapses by a previous approach in 5.11% of cases (*n* = 24); and in elderly patients over 60 years old in 84.2% of cases (*n* = 128), with the oldest patient operated on by TEP technique being 75 years old (Table 5 and Table 6).

Patients undergoing an open procedure presented significant comorbidities: ischemic heart disease (IHD), a history of acute myocardial infarction (AMI), arrhythmias or ventilatory disorders that contraindicated general anesthesia with oro-tracheal intubation, or were large/incarcerated/strangulated inguinal-scrotal hernias but did not require enteral resections.

There were no conversions from laparoscopic surgery to open surgery. Of the eight TAPP interventions, five were conversions of the TEP procedure. The reasons for the conversion (or rather, the change in operative tactics) were the lack of assessment of the size of the hernia in obese patients or the incarcerated hernia that could not be reduced after anesthetic relaxation.

The duration of hospitalization was as follows: average duration for TEP/TAPP—1 day, for the Lichtenstein procedure—2.54 days.

The duration of the surgical intervention in the case of TEP was 55 min for the unilateral approach and 90 min for the bilateral approach. In the case of TAPP, the average duration was 70 min for unilateral hernia and 100 min for bilateral hernia. The classic procedure (Lichtenstein) had a duration of 95 min for unilateral hernias and 130 min for bilateral hernias.

Simultaneous interventions, only for laparoscopic surgeries, consisted of the following:

*n* = 7 cases (1.41%)—umbilical hernias–alloplastic procedure;

*n* = 3 cases (0.60%)—laparoscopic cholecystectomies;

*n* = 2 cases (0.40%)—PPH (Longo stapled hemorrhoidopexy);

*n* = 2 cases (0.40%)—postoperative hernia—alloplastic procedure;

*n* = 5 cases (1.01%)—exploratory laparoscopy in order to control the reduced intestinal loop.

A 15 × 12 cm polypropylene mesh was introduced with 10 mm trocar and placed after unrolling it in preperitoneal space. All potential hernia sites were covered. The mesh was placed from the pubic symphysis (overlapping 2 cm to the opposite) to the anterior superior iliac spine laterally. Mesh fixation was performed with Sorbafix.

Mesh fixation in laparoscopic interventions was performed in 100% of bilateral inguinal hernias operated in TEP and in 78% (*n* = 190) of unilateral TEP in all TAPP interventions. In bilateral TEP, the drain tube was inserted in 82% of cases (*n* = 199), and in 23% of cases (*n* = 52) in unilateral TEP.

The placement of the Redon suction drain tube was necessary in 81.2% of the total of 16 cases and in TAPP bilateral procedures; 25% of cases (*n* = 2 cases) in TAPP unilateral; and in most of the open interventions, *n* = 13 cases.

Regarding complications, there were no intraoperative complications recorded in any procedure, while postoperative complications were found in 10.23% of the cases (*n* = 48) in the extraperitoneal approach, such as seroma, urinary retention, hematoma, postoperatively, and recurrences after the TEP approach found in 0.42% of cases (*n* = 2). Reintervention was performed in 1.41% of patients (*n* = 7) with complications (Table 7).

Five patients operated upon with the TEP technique (1.07% of cases) underwent laparoscopic surgery for extraperitoneal hematoma, parietal hematoma, or hemoperitoneum, and thus the hematoma evacuation and the hemostasis control were performed. In two patients with open hernia repair, surgical intervention was performed to control hemostasis and evacuate the hematoma. One case also underwent a unilateral orchiectomy (Table 7).

## 4. Discussion

In this study, our goal was to demonstrate that the TEP approach to hernia repair is an optimal22 solution in properly selected cases.

We used TEP as a routine technique for hernia repair, and even though the indications for TEP and TAPP overlapped to a large extent, we preferred to choose the TEP technique in those cases.

Even though the design of the study was not ideal (the number of patients operated upon with the TEP technique was significantly higher than TAPP or the classic technique), we concluded that if a correct indication can be established, then this type of surgical approach can be beneficial for both the patient and the surgeon.

We had a total of 493 consecutive patients diagnosed and operated on for inguinal hernia; in 469 cases (95.1%), we performed TEP with good postoperative results.

In our study, the indications for the TEP technique were patients with reducible hernia, incarcerated hernia, recurrent hernia after the classic procedure, ASA score 1–3, history of intra-abdominal infections, and those without previous surgical interventions in the lower abdomen.

In five cases of TEP indication, we had to switch to the TAPP surgical procedure; this was not necessarily a conversion, but a lack of preoperative assessment of the size and type of hernia (we discovered after anesthesia that the hernia was larger than expected in obese patients, with an incarcerated intestinal loop that could not be reduced).

A significant difference in the two laparoscopy groups was the average duration of the operation (significantly shorter in the TEP group, 50/90 min for uni/bilateral hernia compared to 70/100 min for uni/bilateral TAPP).

There are studies in the literature reporting a longer operating time for laparoscopic hernia compared to open repair. Some studies have reported a similar operative time in open and laparoscopic hernia repair. The learning curve for laparoscopic hernia surgery is longer, as reported in the literature [3,14,15,16]. In our study, the average duration of the operation in the TEP group was shorter than in the other two groups.

The main intraoperative complication in laparoscopic TEP surgery described in the literature is the accidental creation of pneumoperitoneum, but in our study, we did not have this complication [15].

A higher level of pain has been reported in previous studies after open hernia repair, as well as in procedures such as Lichtenstein hernioplasty, compared to laparoscopic hernia repair (TEP or TAPP) [14,15,17].

In our study, at the 10 days follow-up of the patients, the pain level was significantly higher in the laparoscopic TEP (in 6 cases) than in the Lichtenstein or TAPP procedure, with no correlation with the findings reported by Courtney et al. [15,18,19,20,21,22]. This may be explained by the numerical difference between the groups with a significantly higher number of patients in the TEP group.

The literature shows debatable reports regarding chronic pain. The metanalyses by Karthikesalimgam et al. [23] and Dhankhar et al. [24,25] reported no significant difference in chronic pain in laparoscopic and open mesh repair [15,16,18,22,23,24,26]. Several other studies have reported a higher incidence of chronic groin pain in open repair patients compared to laparoscopic repair patients [17,27,28].

Persistent chronic groin pain affects patients’ quality of life [16,19,20,21,22,23,24,26]. The possible reason for postoperative chronic pain in open hernia mesh repair is unclear. It may be due to nerve injury or nerve entrapment, the quality of the mesh used in the repair, or perhaps improper positioning of the mesh in the inguinal canal. However, we need to consider that many patients may report groin discomfort as chronic groin pain [25,29]. In our study, we had only one patient who had persistent groin pain up to one month after Lichtenstein surgery.

Postoperative complications were found in 10.23% of cases of the extraperitoneal approach (such as seroma, urinary retention, hematoma, postoperative algic syndrome); among those patients, in 29 cases we had urinary retention, a significantly high number, because we did not perform a routine preoperative bladder survey.

This complication does not correspond to the specialized literature in which urinary retention in the TEP procedure has a low incidence.

There were patients who had other complications (scrotal hematoma, intra/extraperitoneal bleeding, pain, or transitory neurological disorders), and most of them required reintervention.

The occurrence of postoperative complications such as wound seroma or infection, scrotal hematoma, and intraperitoneal/extraperitoneal bleeding were not statistically significant in our study.

Postoperative pain had a low incidence, being encountered in six cases after the TEP approach, with this complication being described with an increased incidence after open surgery.

The recurrence rate was significantly low in the TEP group, 0.42%, according to data cited in the literature [1,4]. There are controversial reports considering the recurrences seen in open and laparoscopic hernia repair. In our study, two recurrences (0.42%) were seen in the laparoscopic TEP group, and no recurrence in laparoscopic TAPP or open hernia repair. This was not in accordance with the study of Baris et al. [24,25,26,27,28,29,30,31,32,33,34], who showed a higher recurrence rate in open mesh repair than in the laparoscopic TEP or TAPP group.

Various studies, meta-analyses, and trial sequential analyses reported no difference in recurrence rates between laparoscopic and open inguinal hernia repair techniques [33,34]. For correctly selected cases, TEP hernia surgery (Figure 1) can be considered a feasible and safe approach [6,8,11,35,36], with minimal postoperative complications, a very low incidence of recurrences, minimal risk of cavitary organs injury, lack of postoperative intra-abdominal adhesions, and not requiring the closure of the peritoneal defect as in TAPP [2,35,36,37,38,39,40]. The technique allows for the conversion at any time to TAPP or open surgery and is ideal not only in bilateral forms but also in recurrences of traditional procedures. Although it requires a long period of training and mandatory knowledge of the other two types of approaches, the technique can become the main option, and in time, a routine for any surgeon with competence in advanced laparoscopic procedures [1,9,39].

All the advantages listed above lead to the conclusion that the TEP technique meets the conditions of an ideal technique for correctly selected cases.

Simultaneous interventions were feasible in the laparoscopic approach, without particular additional risks [1,3,4,7]. Laparoscopy versus the open approach is a clear, uncontroversial topic (Figure 2).

## 5. Conclusions

The significant conclusions that can be drawn from the TEP technique are that it is a procedure that fulfills all the current requirements of minimally invasive surgery, it is a safe and feasible procedure in cases of recurrences in elderly patients who are correctly selected and prepared, and has reduced perioperative complications.

Although there is no ideal operative technique for the inguinal hernia approach, laparoscopic techniques have significant benefits for the patient. Finally, it may be a preference of the operating surgeon for TEP or TAPP, and we have demonstrated that TEP can be suitable in the treatment of inguinal hernias.

In addition, this technique is a useful approach for bilateral hernias, did not supplement the operative risk, and did not extend the duration of hospitalization.

In primary unilateral inguinal hernia and in bilateral hernia cases, the laparo-endoscopic approach (TEP, transabdominal preperitoneal) is the first choice, provided by surgeons with sufficient expertise (as part of the International Guidelines of various hernia societies) [14,15].

In our study, TEP was associated with a low recurrence rate and postoperative pain, but the most important advantage compared to TAPP was represented by the shorter duration of the operative time in unilateral or bilateral hernia. That is why we recommend it as a method of choice in correctly selected cases.

## Figures and Tables

**Figure 1 jcm-11-05652-f001:**
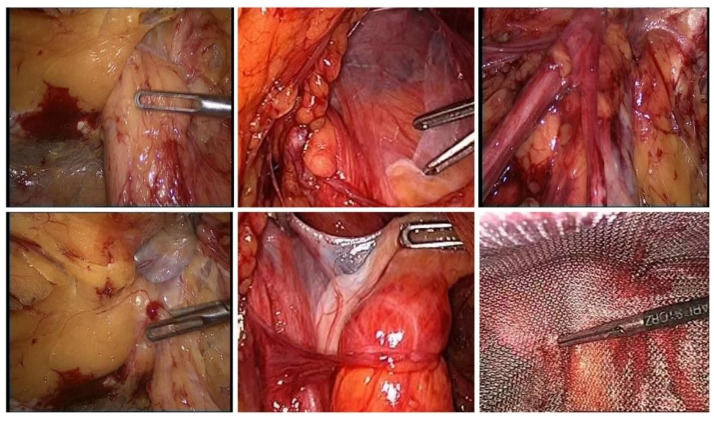
Intraoperative aspects during hernia surgery.

**Figure 2 jcm-11-05652-f002:**
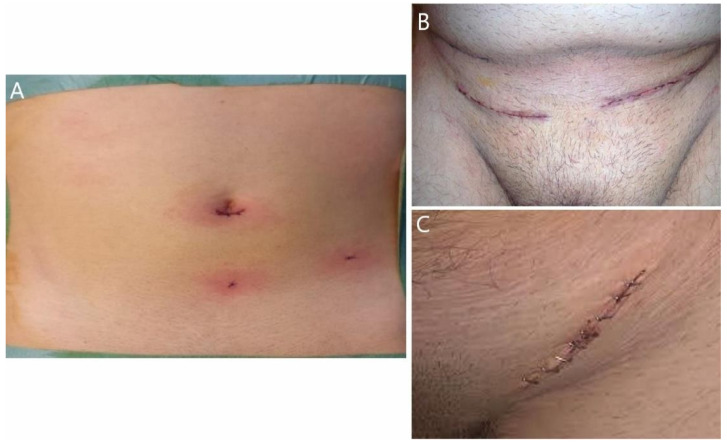
(**A**) Laparoscopy procedure; (**B**,**C**) open approach procedure.

**Table 1 jcm-11-05652-t001:** Type of surgery and the number of patients.

Type of Surgery	Number of Patients	Percentage
Unilateral TEP	243	49.2
Bilateral TEP	226	45.84
Unilateral TAPP	5	1.01
Bilateral TAPP	3	0.6
Lichtenstein “tension free”—bilateral	3	0.60
Lichtenstein “tension free”—unilateral	13	2.63
Total	493	100.0

**Table 2 jcm-11-05652-t002:** The average age of the patients according to the type of operation performed.

Descriptives			
	Type of Surgery		Statistic
AGE	TEP unilateral	Mean	47.63
		Median	44
		Std. deviation	14.98
	TEP bilateral	Mean	48.83
		Median	46
		Std. deviation	13.959

**Table 3 jcm-11-05652-t003:** Number of elderly patients according to the type of surgery.

Type of Surgery	Valid		
	*n* = 152	% of Elderly Patients	% Total (493)
Unilateral TEP	46	30.2	9.33
Bilateral TEP	82	63.2	16.6
Unilateral TAPP	5	3.28	1.01
Bilateral TAPP	3	1.97	0.60
Lichtenstein “tension free“—bilateral	3	1.97	0.60
Lichtenstein “tension free“—unilateral	13	8.55	2.63

**Table 4 jcm-11-05652-t004:** Type of surgery performed according to the patient’s gender.

Type of Surgery * Sex Crosstabulation					
			Gender		Total (*n* = 493)
			Female	Male	
Type of surgery	Uni/bilateral TEP	Count	55	414	469
		% of Total	11.15%	83.97%	95.13%
	Uni/bilateral TAPP	Count	2	6	8
		% of Total	0.40%	1.21%	1.62%
	Lichtenstein “tension free”—uni/bilateral	Count	3	13	16
		% of Total	0.60%	2.63%	37.20%
**Total**		Count	60	433	493
		% of Total	12.17%	87.83%	100%

**Table 5 jcm-11-05652-t005:** The type of hernia and laparoscopy used as a surgical solution (1).

		TEP	TAPP
**Type of hernia**	Indirect hernia—NYHUS I,II	28	4
	Direct hernia—NYHUS III A	210	2
	Complex hernia—NYHUS IIIB	205	1
	Femoral hernia—NYHUS IIIC	2	0
	Recurrent hernias NYHUS IV	24	1
	Total	469	8

**Table 6 jcm-11-05652-t006:** The type of hernia and laparoscopy used as a surgical solution (2).

		Reductible	Incarcerated	Strangulated	Total
**Surgery type**	TEP	249	220	0	469
	TAPP	3	5	0	8
	Lichtenstein	3	10	3	16
	Total	255 (51.72%)	235 (47.66%)	3 (0.60%)	493

**Table 7 jcm-11-05652-t007:** Perioperative complications according to the surgical procedure.

Type of Complication/Surgical Procedure		TEP: 469	TAPP: 8	Lichtenstein: 16
**Intraoperative Complications**		0	0	0
Immediate postoperative complications	Wound seroma	3 (0.63%)	0	2 (12.5%)
	Urinary retention	29 (6.18%)	1 (12.5%)	1 (6.25%)
	Wound/inguinal—scrotal hematoma	3 (0.63%)	1 (12.5%)	2 (12.5%)
	Intra/extraperitoneal bleeding	5 (1.07%)	0	0
	Suppurative complications/mesh infection	0	0	0
	Pain/transitory neurological disorders	6 (1.27%)	0	2 (12.5%)
Late postoperative complications	Pain/persistent neurological disorders	0	0	1 (6.25%)
	Recurrences	2 (0.42%)	0	0
Total		48 (10.23%)	2 (25%)	8 (50%)

## Data Availability

Not applicable.

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
