# Peer review of "Extraperitoneal Laparoscopic Approach in Inguinal Hernia—The Ideal Solution?"

_jcm, 2022, doi:10.3390/jcm11195652_

Round 1

Reviewer 1 Report (Previous Reviewer 1)

I thank the authors for the changes made

Author Response

Thank you once again for your pertinent observations and appreciation of our work.

Reviewer 2 Report (New Reviewer)

Interesting manuscript that is worth to publish.Unfortunately ,as presented,it is not attractive.This methodology is certainly feasible as it was selected for the operations.The authors must be more enthusiatic :it is not appropriate to raise question marks.The operation technique is well described  but the tables are confusing and shall be improved;the quality of the photos is not disputable.The constructions of the  hernia meshes selected are not given.The first paragraph of the discussion is awkward;It is clear that there is no consensus,This discussion must refer to additionnal publications to give a good support to this manuscript ; the conclusion shall give a clear cut opinion.The bibliography is not sufficient and shall be presented according to the guidelines provided by the editor.

Author Response

Thank you once again for your pertinent observations and appreciation of our work.

We added 26 more bibliographical references, including studies and meta-analyses, and in the Discussion section we compared the results of our study with these publications. In this way we managed to highlight the advantages of the TEP technique which was the main operative technique used in our study. We also had shortcomings in this study, mainly the design of the groups, but this did not  essentially influenced the results obtained.

We deleted the first paragraph of the Discussion and we focused on describing our results by comparing them with data from the literature.

We have also provided a description of the mesh used and of the fixation technique.

Regarding the tables, our intention was to use as much information as possible (compressed into a small space). The more complex tables are no. 4 (sex crosstabulation) and no. 7 (postoperative complications). We have provided text explanations below each table with the relevant information. We benefited from the help of a statistician who had suggestions on how the tables were drawn up. If you still consider (after this explanation) that the tables are too crowded or confusing, we will replace them, but we consider that we may probably lose information by doing so.

And finally, we tried to translate our enthusiasm for the method into the right words.

Round 2

Reviewer 2 Report (New Reviewer)

   Good revision

This manuscript is a resubmission of an earlier submission. The following is a list of the peer review reports and author responses from that submission.

Round 1

Reviewer 1 Report

The manuscript is very interesting and original. This comparison of techniques is useful. I recommend integrating the statistical analysis and making it more detailed

Author Response

Thank you for the attention given to the article “Extraperitoneal laparoscopic approach in inguinal hernia – the ideal solution?”

Comments coming from the reviewers made us understand the importance of discussion and highlighting the possible advantages in TEP procedure. It is the result of our work and it is very important for us. Thank you for your appreciation.

We read all the comments carefully and responded to each one.

We came up with more detailed discussions based on our statistical analysis and compared our results with the literature.

Reviewer 2 Report

The authors have reported a single centre, single team, case-control study of open versus TAP versus TEP inguinal hernia repair. Based on the clinic’s surgical experience in hernia pathology, the study aimed to assess feasibility of the extraperitoneal approach in comparison to conventional procedures. I commend their efforts and the significant clinical workload carried out by the team. Below I have commented on individual sections and highlighted areas of weakness, which I hope the authors will act on.

Methods

Describe a significant number of hernias by a single team. Methodological patient selection, definitions provided and workup seem sound. Hernia diagnostic criteria provided clearly which is positive. Unfortunately, indication for drain is not specified which does not allow comparison. The learning curve of each surgeon (e.g. years of operative experience with open and then laparoscopic hernia surgery) is not reported, making it difficult to assess the potential learning curve or translatability into other centres. Reasons for conversion of TEP to TAPP not outlined or explained so not able to assess patient selection. 

Specific points:

63 - postoperative control. Control of what post-operatively?

66-67 - TAPP was performed in the case of previous interventions by sub-umbilical approach or 66 conversion of the TEP procedure. Indications for TEP and TAPP are different in this series, so including TAPP when it is a conversion of TEP introduces bias, unless the stage of the operation at which conversion occurs is outlined for clarity of comparison, which it is not.

71 - median should read medial.

Results

Generally demonstrate large number (469) of TEP patients. Unfortunately, only age and sex demographics presented, with no clarification of past medical history, ASA score, Charlson Comorbidity Index score, or equivalent to allow comparison between patients; this makes it difficult to accurately assess the patient populations who are appropriate for each technique. Complication rates recorded are in keeping with other literature on the topic. 

Specific points:

89-92 - Figure 2. This is a figure. Schemes follow another format. If there are multiple panels, they should be listed as: (a) Description of what is contained in the first panel; (b) Description of what is contained in the second panel. Figures should be placed in the main text near to the first time they are cited. A caption on a single line should be centered. This should not be included in the main text.

129-130: Of the 8 TAPP interventions, 5 were conversions of the TEP procedure. Therefore, these TEP should be classified as failed if following an intention to treat analysis and not included in the TAP arm, as indication for TEP and TAP was different, therefore, not comparable. 

Discussion

The discussion provides statements to the use of laparoscopy and the different techniques, but does not draw on the data reported sufficiently. Because of the lack of patient demographics or reasoning behind TEP to TAPP conversion in the results, these are not discussed in the discussion and therefore there is no true description of the indications or limitations within the current dataset, rather just repetition of the literature. This leaves us with a literature review and discussion of very few new data from the author's centre, which does not add a new perspective. 

Author Response

Thank you for the attention given to the article “Extraperitoneal laparoscopic approach in inguinal hernia – the ideal solution?”

Comments coming from the reviewers made us understand the importance of discussion and highlighting the possible advantages in TEP procedure. It is the result of our work and it is very important for us. Therefore, we thank you for your in-depth comments. They were very useful.

We corrected the mistakes – we deleted that paragraph (lines 89-92), replaced the median with the medial, replaced the number of cases (313) with 493 (the correct one).

In Methods we explained in detail the indications for each technique (TEP, TAPP and classic), we introduced information about the surgical team experience. In the matter of conversion it was our mistake because we did not explain correctly – we hope that now we have managed to explain what happened. In fact, there was no real conversion (for reasons such as peritoneal rupture), those 5 cases of TEP indication (which we stated in the TAPP conversion) were in fact cases of changing the operative tactics, with the lack of preoperative assessment of hernia size (in obese patients), or cases in which the incarcerated hernia could not be reduced (after anesthesia and relaxation). This is why we consider those cases in the TAPP group (the preoperative indication was not accurate).

In the matter of postoperative control – it is a routine follow –up, at 1,5 weeks after surgery, regardless of the type of procedure applied. At that time we remove the stitches and clinically assess the appearance of the wound (for seroma, hematoma, etc). We introduced this information in the text.

We also added information about ASA score.

In Discussion – we extended the analysis in our groups, highlighting the advantages and indications of TEP or TAPP techniques, as it resulted from our study. We also compared the results with the literature.

Reviewer 3 Report

In this study the investigators tried to compare the TEP and TAPP laparoscopic approach for repairing inguinal hernias. As a conception is very interesting, based on the ongoing debate about which method is more feasible. However, the aim of comparing the two methods was not fulfilled. 

Abstract:

The word "eventually" in line 20, should be deleted because it gives an arrogant style.

The abbreviations of TAP and TEPP should be explained

Introduction: 

Line 36: "800.000 inguinal...." should be stated with words and not number, because it is the beginning of a new sentence. 

Materials and Methods:

line 58: stating the number of the patients including should be in the Result section. Moreover, the authors reported that 313 patients (and not 493 as the write in line 58) were included who underwent 493 operations.

No comparison was made between the two groups and no statistical method was described.

The authors did not describe the TAPP and Lichtenstein method in detail, as they have properly done with the TEPP one.

No inclusion/exclusion criteria were described on which the investigators based to form the sample.

Results:

The authors analyzed retrospectively 313 patients, undergoing totally 493 operations (Lichtenstein, TEPP, TAP). Three groups were formed. 1) Lichtenstein (n=16), 2) TAPP (n=8) and 3) TEPP (n=469). However, they stated 493 patients in line 81. In addition, the three groups are completely different regarding the number of the patients included and any comparison would be definitely biased.

Lines 89-92 should be deleted. 

The authors presented the age of each group as means in Table 2. However, they did not explain why they use mean and not median and they did not provide SDs as well.

Line 97: (N=414 operations) and not patients. the number of the patients are 313.

Poor quality of the data presented in Tables.

Discussion:

No discussion was presented. They did not discuss thew pros and cos of every method. They did not compare their results with respective results in the bibliography. They did not state strengths and limitations of their study. The discussion should be much more detailed and should also be expanded.

Conclusion:

The authors concluded at a kind of superiority of TEP, but this conclusion cannot be drawn by the data and results of this study.

Author Response

Thank you for the attention given to the article “Extraperitoneal laparoscopic approach in inguinal hernia – the ideal solution?”

Comments coming from the reviewers made us understand the importance of discussion and highlighting the possible advantages in TEP procedure. It is the result of our work and it is very important for us. Therefore, we thank you for your time and attention and for your very pertinent comments.

We deleted the word “eventually”, explained the abbreviations TAPP and TEP, the phrase that had a number at the beginning (800 000) was also changed.

We have described in detail the Lichtenstein and TAPP techniques. We also specified the inclusion criteria for each group and the statistical tools used.

We corrected the number 313 – it was a transcription error. The correct number of patients is 493.

We deleted the lines 89-92 – it was also an unfortunate mistake.

In Discussions we completed the analysis with several data from our study, including the indications for each method, the results, the benefits comparing them with the data from the literature.

Conclusions – we have nuanced the information, we have now based our conclusions on the results of the personal study. We emphasized the advantages and superiority of TEP in the correctly selected case.

Round 2

Reviewer 3 Report

I congratulate the authors who have made a great effort to improve their manuscript. They described briefly every technique applied and reported the inclusion criteria. They have also added information about the operation time needed and expanded the discussion section. However, some major issues that should be improved.

Firstly, once again it remains very confusing when the authors use the number for the operations (493) instead of the number for the patients (313) and vice versa (lines 60, 142)

Nevertheless, the most critical pitfall of this study is that there is not any type of comparison between the three groups which would let us conclude the non-inferiority of TEP vs the other techniques. We are provided only with the descriptive statistics of the sample and therefore any conclusion about the differences between the three methods would be misleading. Moreover, based on the fact that the three groups differ significantly regarding the number of patients included, it is more than difficult to make any comparisons.

Additionally, table 2 is just a copy of the SPSS output and it is quite confusing when studying it. Finally, the authors did not provide us with the information about the skewness of the data distribution, to evaluate the mean or the median of the sample.
